# Antioxidant Potential and Cytotoxic Effect of Isoflavones Extract from Thai Fermented Soybean (Thua-Nao)

**DOI:** 10.3390/molecules26247432

**Published:** 2021-12-08

**Authors:** Kanokwan Kulprachakarn, Supakit Chaipoot, Rewat Phongphisutthinant, Narisara Paradee, Adchara Prommaban, Sakaewan Ounjaijean, Kittipan Rerkasem, Wason Parklak, Kanittha Prakit, Banthita Saengsitthisak, Nittaya Chansiw, Kanjana Pangjit, Kongsak Boonyapranai

**Affiliations:** 1School of Health Sciences Research, Research Institute for Health Sciences, Chiang Mai University, Chiang Mai 50200, Thailand; sakaewan.o@cmu.ac.th; 2Environmental and Occupational Health Sciences and Non Communicable Diseases Research Group (EOHS and NCD Research Group), Research Institute for Health Sciences, Chiang Mai University, Chiang Mai 50200, Thailand; kittipan@rihes.org (K.R.); wason.p@cmu.ac.th (W.P.); 3Science and Technology Research Institute, Chiang Mai University, Chiang Mai 50200, Thailand; supakit.ch@cmu.ac.th (S.C.); rewat.p@cmu.ac.th (R.P.); 4Center of Excellent in Microbial Diversity and Sustainable Utilization, Faculty of Science, Chiang Mai University, Chiang Mai 50200, Thailand; 5Oxidative Stress Cluster, Department of Biochemistry, Faculty of Medicine, Chiang Mai University, Chiang Mai 50200, Thailand; narisara.p@cmu.ac.th (N.P.); adchara.p@cmu.ac.th (A.P.); 6Department of Surgery, Faculty of Medicine, Chiang Mai University, Chiang Mai 50200, Thailand; 7Faculty of Pharmacy, Payap University, Chiang Mai 50000, Thailand; kanittha_puprakit@hotmail.com (K.P.); banthita_s@payap.ac.th (B.S.); 8School of Medicine, Mae Fah Luang University, Chiang Rai 57100, Thailand; nittaya.cha@mfu.ac.th; 9College of Medicine and Public Health, Ubon Ratchathani University, Ubon Ratchathani 34190, Thailand; kanjana.pa@ubu.ac.th

**Keywords:** antioxidant, cytotoxicity, Thai fermented soybean, Thua-nao, *Bacillus subtilis* var. *Thua-nao*

## Abstract

Thua-nao, or Thai fermented soybeans, is a traditional Lanna fermented food in Northern Thailand. It is produced by using a specific bacterial species called *Bacillus subtilis* var. *Thua-nao*. We investigated the antioxidant activity and cytotoxic effect of isoflavones from Thua-nao. The phenolic compound contents and total flavonoid contents were determined by spectrophotometry. The antioxidant activity was examined using the ABTS, FRAP, and DPPH assays. The isoflavone contents and phenolic compositions were examined by the high-performance liquid chromatography (HPLC) and liquid chromatography-mass spectrometry (LC-MS) techniques. The ability of isoflavones to inhibit human cancer cell growth was assessed by the MTT assay. The total phenolic content, total flavonoid content, and antioxidant activities of the isoflavones were 49.00 ± 0.51 mg GAE/g of dry extract (DE), 10.76 ± 0.82 mg QE/g of DE, 61.03 ± 0.97 µmol Trolox/g of DE, 66.54 ± 3.97 µM FeSO_4_/g of DE, and 22.47 ± 1.92% of DPPH inhibition, respectively. Additionally, the isoflavone extracts from Thua-nao had high isoflavone contents and polyphenolic compound compositions, especially daidzein and genistein. The isoflavone demonstrated a weak inhibition of MCF-7 and HEK293 cancer cell growth. It has a high antioxidant component, which is beneficial and can be developed for new therapeutic uses. However, further studies on the benefits of Thua-nao should be performed for realizing better and more effective uses soon.

## 1. Introduction

Thua-nao is an alkaline fermented food condiment produced from soybeans that are native to Thailand. Alkaline fermented soybean-based products are generally consumed in Asian countries like Japan, China, Korea, and Indonesia. Thai fermented soybeans or Thua-nao are a traditional Lanna fermented food in Northern Thailand. It is a favorite food consumed among Northern Thai people (i.e., Chiang Rai, Chiang Mai, Mae Hong Son, Lamphun, and Lampang) [1]. It is produced by using a specific bacterial species called *Bacillus subtilis* var. *Thua-nao*, which can create protein-digesting enzymes in soybeans, giving the beans their texture, unique smell, and taste. Thua-nao is produced for household consumption, preserved for a longer shelf life, allowing people in the community to have a self-sufficient lifestyle. It is often a part of every household for cooking, like shrimp paste in the central region, added to vegetable soup, wrapped in banana leaves, or steamed or grilled enough to eat with sticky rice. The appearance of Thua-nao is similar to that of Japan’s natto products. Other countries’ traditional dishes that resemble Thua-nao are kinema, a traditional dish of Nepalese and Indian people, and chungkookjang, used as a seasoning in Korea. Presently, vegetables and fruits are consumed to provide the body with essential nutrients, almost all of which are biologically active compounds that can reduce the risk of serious diseases, such as diabetes, cardiovascular disease, and cancer [2].

Secondary plant biological chemicals known as isoflavones are found in soybeans, particularly in the soy germ. Isoflavones are classified as estrogen-like active substances, which are classified as phytoestrogens. By inhibiting estrogen’s activity in humans, it can help reduce the risk of coronary artery disease and breast cancer. Isoflavones are also shown to be compounds that diminish lipid adhesion in the arteries [3], lowering the risk of osteoporosis [4], and osteoporosis prevention via increasing bone formation and inhibiting bone resorption, as well as lowering postmenopausal women’s adverse symptoms. In addition, isoflavones also reduce the risk of prostate cancer in elderly men and reduce the activity of carcinogens such as breast cancer and prostate cancer as well. Thua-nao is highly nutritious, containing proteins, carbohydrates, fat, vitamins, and other minerals, such as iron and calcium, as well as the essential compounds, isoflavone aglycones (daidzein, genistein, and glycitein), phytochemical substances with health-promoting effects. A nutritious substance with antioxidant activity, phytoestrogen, which works similarly to the female hormone estrogen (17ß-estradiol E2 isoflavones), can be used to alleviate postmenopausal symptoms or may have a preventative or modulating effect on physiological disorders or postmenopausal symptoms, as well as diseases like cancer, osteoporosis, Alzheimer’s disease, obesity, and cardiovascular disease [5]. Many studies have revealed that the high-performance liquid chromatography (HPLC) method using a C-18 column with UV detection in the optimum condition was performed to identify and quantify the isoflavone contents in extracts of soybeans [6,7]. Furthermore, various isoflavone forms were identified and confirmed by liquid chromatography-mass spectrophotometry (LC-MS) [8].

Soybean isoflavones are generally in the form of glucosides, acetyl glucoside, and malonyl glucosides and consist of dicin, genistin, glycetin, acetyldisin, acetylgenistin, acetylglycetin, malonyldisin, malonylgenistin, and malonyl glycetin, which is a structure with a large molecular size, resulting in being absorbed rather slowly in the human body. Previous studies [9,10] have demonstrated that the enzyme ß-glucosidase from Bacillus coagulant PR03 could convert isoflavones from glucoside forms, acetyl glucoside, and malonyl glucoside into aglycone, which contains daidzein, genistein, and glycitein. Isoflavone aglycones are more attractive than glucoside structures, because they can be absorbed efficiently into the small intestine of humans and have a more substantial estrogen-like effect than isoflavone glucosides, based on the properties of isoflavones mentioned above. Therefore, the present study is to determine the antioxidant activity and cytotoxic effect of ethanolic isoflavone extracts from Thai fermented soybeans (Thua-nao).

## 2. Results

### 2.1. Total Phenolic Content

The total phenolic content, expressed as the mg gallic acid equivalent (GAE) per gram of dry extract, is shown in Table 1. The amount of phenolic content of the ethanolic isoflavone extracts from Thua-nao was found to be 49.00 ± 0.51 mg gallic acid equivalent (GAE) per gram of dry extract.

### 2.2. Total Flavonoid Content

The total flavonoid content expressed as the mg quercetin equivalent (QE) per gram of dry extract is shown in Table 1. The total flavonoid content of the ethanolic isoflavone extracts from Thua-nao was found to be 10.76 ± 0.82 mg quercetin equivalent (QE) per gram of dry extract.

### 2.3. Antioxidant Activity

The antioxidant activities were determined by the modified ABTS radical scavenging, ferric-reducing antioxidant power (FRAP), and DPPH radical scavenging assays. As shown in Table 1, the level of antioxidant capacity of the ethanolic isoflavone extracts from Thua-nao was 61.03 ± 0.97 μmol Trolox per gram of dry extract and 66.54 ± 3.97 μM FeSO_4_ per gram of dry extract, and the percentage inhibition of the DPPH radical by the extract was 22.47 ± 1.92% at the concentration of 5 mg/mL, respectively.

### 2.4. Isoflavone and Polyphenolic Compositions

The analysis of isoflavone aglycones (genistein and daidzein) and glycones (genistin and daidzin) was performed by HPLC and LC-MS. Table 2 shows the compositions and contents of the isoflavones and polyphenolic compounds in the ethanolic isoflavone extracts from Thua-nao and the levels of the isoflavone contents such as daidzein, genistein, genistin, and daidzin, respectively (Appendix A). It was found that the amount of aglycone isoflavones was higher than aglycone isoflavones, especially daidzein (632.27 ± 1.11 mg per kg of dry extract). Furthermore, eight polyphenolic compounds (mainly rutin, isoquercetin, catechin, tannic acid, quercetin, gallic acid, eriodictyol, and apigenin, respectively) were also found in the extracts (Appendix A).

### 2.5. Cytotoxicity in Various Cancer Cells

In this study, the cytotoxic activity of the ethanolic isoflavone extract from Thua-nao was determined using the MTT assay in a human hepatocellular carcinoma cell line (HepG2), human embryonic kidney cell line (HEK293), and human breast adenocarcinoma cell line (MCF-7), which were exposed to 0–800 µg/mL of the extract and 0–200 μM of genistein and daidzein (as a positive control) at two incubation periods of 24 and 48 h, respectively. The 50% inhibitory concentration (IC_50_) value, which refers to the concentration of the extract that could inhibit 50% of cell growth, was determined as shown in Figure 1, Figure 2 and Figure 3 and Table 3. In Figure 1, the extract decreased the amount of viable HepG2 cells with an IC_50_ value of 722 μg/mL and 762 μg/mL after 24 and 48 h of treatments, respectively. HepG2 cells treated with genistein at 48 h exhibited the most cytotoxic activity at a IC_50_ value of 112 μg/mL (Figure 1B). The most severe effect was seen after 48 h of treatment, in which the extract decreased the amount of viable HEK293 cells in a concentration-dependent manner, and the 50% of cell growth inhibition value was 392 µg/mL (Figure 2A). It was demonstrated that the extract killed 50% of the cells (IC_50_) at a lower concentration at the 48-h incubation period when compared to the other cancer cells tested. In Figure 3, the extract decreased the cell viability in a dose-dependent manner, and the IC_50_ value of the extract was found to be 698 μg/mL after 24 h of incubation in MCF-7 cells. The treatment with genistein showed the most cytotoxic effect in the HEK293 cells (Figure 2B) and MCF-7 cells (Figure 3B) at both incubation periods as well. The ethanolic isoflavone extracts from Thua-nao tended to inhibit these three types of cancer cell line growths.

IC_50_: 50% inhibitory concentration; HepG2: human hepatocellular carcinoma cell line; HEK293: human embryonic kidney cell; MCF-7: human breast adenocarcinoma cell line.

## 3. Discussion

Thua-nao is a traditional fermented soybean that is popular among Thai people living in the north. The consumption of soy-fermented foods has been found to provide several health benefits, including being a good source of protein and containing soy-isoflavones, which are well-known for their pharmacological and antioxidant activities [11]. Isoflavone is a soy phytoestrogen and a biologically active component of several agriculturally important legumes, such as soy, peanuts, green peas, chickpeas, and alfalfa. Isoflavones are a subclass of flavonoids; they are the major phytoestrogens naturally found in plants. Isoflavones are mainly present in soybeans (5–30 mg/100 g). Isoflavones are often found as glycosides. Daidzin, genistin, formononetin, and biochanin A are the most common isoflavones. The fermentation or digestion of soybeans or soy foods releases a sugar moiety from the isoflavones glycosides, genistin, daidzin, and glycitin, which are then transformed to the isoflavone aglycones, genistein, daidzein, and glycitein. Isoflavones have been associated with several health benefits. Isoflavones consumed at levels found in soy foods can help maintain blood vessel health, relieve menopausal symptoms, lower risks of breast cancer, and lower cholesterol and glucose levels [12]. Foods containing isoflavones may assist in maintaining cellular health by increasing the body’s amount of antioxidants. Based on the results of this study, the ethanolic isoflavone extract from Thua-nao exhibited antioxidant and free radical scavenging properties. It revealed high total phenolic contents, total flavonoid contents, and antioxidant activity. Our results were similar to several studies that reported many fermented soybean foods exhibiting high contents of antioxidative agents compared to unfermented soybeans [13]. The antioxidant and anti-inflammatory characteristics of isoflavones might also contribute to protection from breast cancer [14]. Some studies have been reported and documented on various kinds of fermented soybeans, such as Japanese natto [15], Korean chungkookjang [16], and Indian kinema [17]. *Bacillus* species, particularly *Bacillus subtilis*, are the most common and have been linked to the fermentation process. A previous study showed that the total phenolic content of Thua-nao extracts varied depending on the source, ranging from 30.46 mg GAE/g extract to 44.58 mg GAE/g extract. The IC_50_ values for the antioxidant activity measured by the DPPH method varied from 2.43 to 3.19 mg/mL of the sample extract, while the total antioxidant activity measured by the carotene linoleate system ranged from 47.21% to 59.45% at 10 mg/mL of the sample extract [18]. According to our findings, the percentage inhibition of the DPPH-free radical scavenging was 22.47 ± 1.92% at a concentration of 5 mg/mL of ethanolic isoflavone extract from Thua-nao. Tempeh, traditional Indonesian food made from fermented soybeans, has the highest concentration of total isoflavones (daidzein and genistein) when compared to other soy products (such as tofu and soy drinks) [19]. Chaiyasut and colleagues [20] found that the highest levels of isoflavone aglycones were 384.30 ± 4.60 and 116.50 ± 1.56 mg/100-g fermented soybeans for daidzein and genistein, respectively. The fermentation time was linked to increases in the isoflavone aglycone concentration and antioxidant activity. The highest antioxidant activity of fermented soybean was found at 24 h of fermentation, with the Trolox equivalent antioxidant capacity (TEAC) 1.98 ± 0.09-µg Trolox/g fermented soybean and a FRAP value of 0.623 ± 0.002-g FeSO_4_/g fermented soybeans. Our study used *Bacillus subtilis* var. *Thua-nao,* a specific bacterial species, for making Thai fermented soybeans. The ethanolic isoflavone extract from Thua-nao displayed antioxidant and free radical scavenging activities, according to our findings. The aglycone derivatives of daidzein and genistein demonstrated the highest isoflavone levels in the extract. According to one study, using a pure starting culture of *Bacillus* species increased the number of aglycones in Thua-nao [21]. Genistein and daidzein are two of the most important active compounds found in soybeans, both of which play a vital role in cancer prevention [22]. Genistein has been shown to inhibit cell growth in a variety of cultured cancer cells, including breast, lung, and prostate cancers, as well as leukemia and lymphoma [14,23]. It affects the cell cycle, apoptosis, angiogenesis, and metastasis at various stages. Remarkably, genistein is considered to have anticancer properties through a variety of mechanisms, including tyrosine-specific protein kinases [5], caspases, B-cell lymphoma 2 (Bcl-2), Bcl-2-associated X protein (Bax), mitogen-activated protein kinase (MAPK), nuclear factor-κB (NF-κB), inhibitor of NF-κB, extracellular signal-regulated kinase 1/2 (ERK 1/2), phosphoinositide 3-kinase/Akt (PI3K/Akt), and the Wingless and integration 1/β-catenin (Wnt/β-catenin) signaling pathway [24]. In this study, we focused on three types of cancer cell lines (HepG2, HEK293, and MCF-7). Our results showed that the isoflavone extracts from Thua-nao have a low inhibitory effect on these cancer cell line growths, but this effect was not significant, However, further studying on the cytotoxicity in various cancer cells, as well as their metabolic antioxidative components, remains to be examined.

## 4. Materials and Methods

### 4.1. Preparation of Isoflavone Extracts from Thua-Nao

The method for isoflavone extraction was modified by Phongphisutthinant and colleagues [10]. Briefly, Thai soybeans were steamed for 3 h, then immediately put in a hot bag until the temperature was at room temperature. Nine percent of *Bacillus subtilis* var. *Thua-nao* was added into the steamed soybeans, the container filled up with air and closed with a cotton stopper, then incubated at 30 °C for 120 h (D-6450, Heraeus, Hanau, Germany). Twenty grams of Thai fermented soybeans were added to 95% of ethanol 100 mL and mixed with a homogenizer for 5 min (MT-30K, MIULAB, Hangzhou, China). After that, they were shakenat a speed of 200 rpm for 2 h (UM-S6060, UMac, Bangkok, Thailand) and then centrifuged at 5000 rpm for 5 min (UNIVERSAL 320R, Hettich, Tuttlingen, Germany). The crude extract was concentrated by evaporating the ethanol solvent under vacuum on a rotary evaporator (R-300, Buchi, Flawil, Switzerland) at a pressure of 175 millibars at 40 °C. Finally, the isoflavones extract was mixed with 1% sodium alginate at a 1:3 ratio, blended with a homogenizer for 5 min, and freeze-dried at −40 °C, 133 × 10^−3^ millibars. Before being used for further analysis, the isoflavone extract was kept in the dark at −20 °C.

### 4.2. Determination of Total Phenolic Content

The total phenolic content of the extract was determined by the Folin–Ciocalteu method [25]. Briefly, 0.5 mL of crude extract was mixed with 2.5 mL of 10% Folin–Ciocalteu reagent for 4 min, followed by the addition of 2 mL of 7.5% (*w/v*) sodium carbonate. The mixture was allowed to stand for 30 min in the dark, and the absorbance was measured at 765 nm (Shimadzu, Kyoto, Japan). The total phenolic content was calculated from the calibration curve, and the results were expressed as mg of gallic acid equivalent (GAE) per gram of dry extract.

### 4.3. Determination of Total Flavonoid Content

The total flavonoid content of the crude extract was determined by the aluminum chloride colorimetric method [26]. In brief, 0.5 mL of crude extract was mixed with 0.1 mL of 10% (*w/v*) aluminum chloride and 0.1 mL of 1-M potassium acetate. Then, 4.3 mL of distilled water was added to the mixture. The mixture was allowed to stand for 30 min, and the absorbance was measured at 415 nm (Shimadzu, Kyoto, Japan). The total flavonoid content was calculated from a calibration curve, and the result was expressed as mg of quercetin equivalent (QE) per gram of dry extract.

### 4.4. Determination of Antioxidant Activity

#### 4.4.1. ABTS Radical Scavenging Assay

Seven millimeters of 2,2′-azino-bis(3-ethylbenzothiazoline-6-sulfonic acid) (ABTS) solution and 2.45-mM potassium persulfate solution were used as stock solutions. The working solution was then prepared by mixing the two stock solutions in equal quantities and allowing them to react for 12–16 h at room temperature in the dark. The solution was then diluted with distilled water to obtain an absorbance of 0.700 ± 0.02 units at 734 nm using a spectrophotometer [27]. Briefly, 10 µL of extract were mixed with 1 mL of the ABTS solution and left in the dark at room temperature for 6 min. The absorbance of the mixture was measured at 734 nm using a spectrophotometer (Shimadzu, Kyoto, Japan). A calibration curve was made by Trolox, and results were expressed as µmol Trolox per gram of dry extract.

#### 4.4.2. FRAP Assay

The Ferric-reducing antioxidant power (FRAP) was determined using the FRAP assay as described in a previous study and slightly modified from Saeio et al. [28]. Briefly, the FRAP reagent was freshly generated by mixing of 300-mM acetate buffer (pH 3.6) and 10-mM 2,4,6 tripyridyl-s-triazine (TPTZ) in 40-mM HCl and 20-mM FeCl_3_ in the proportion of 10:1:1. In this study, 20 µL of the samples were mixed with 180 µL of the FRAP reagent in 96-well plates and incubated for 5 min at room temperature in the dark until the reaction was completed. The reacted samples were measured at 595 nm by using a microplate reader (BioTek Synergy H4 Hybrid Reader, BioTek Instruments Inc., Winooski, VT, USA). Ascorbic acid (1 mg/mL) was used as a positive control. The standard curve was produced by plotting the absorbance at 595 nm versus different concentrations of ferric sulfate (FeSO_4_) ranging from 0 to 1 mM. The FRAP value was calculated from the FeSO_4_ standard curve and expressed as μM FeSO_4_ per gram of dry extract.

#### 4.4.3. DPPH Radical Scavenging Assay

The antioxidant activity was evaluated using 2,2-diphenyl-1-picrylhydrazyl hydrate (DPPH) free radical scavenging activity according to the method modified from Paradee et al. [27]. Briefly, 200 µL of samples were mixed with 200 µL of 0.4-mM DPPH solution and incubated at room temperature for 30 min in the dark. Trolox was used as the positive control. Then, the absorbance was measured at a wavelength of 517 nm using a microplate reader (BioTek Synergy H4 Hybrid Reader, BioTek Instruments Inc., Winooski, VT, USA). The ability of DPPH radical scavenging was expressed as the percentage inhibition and calculated using the following equation:DPPH scavenging activity (%) = (A_0_−A_1_)/A_0_ × 100
where A_0_ is the absorbance of the control, and A_1_ is the absorbance of the sample.

### 4.5. HPLC and LC-MS Analyses for Isoflavone Compositions

The isoflavone contents of the extract from Thua-nao were examined by the high-performance liquid chromatography (HPLC) technique with slight modifications [29]. Briefly, the extract powder (1 g) was extracted with 5 mL of methanol with shaking at 60 rpm in a water bath for 12 h at 37 °C. The extracts were recovered by centrifugation using a centrifuge at 12,000× *g* at 4 °C for 15 min. The sample solution was filtered through a 0.45-µm membrane and then analyzed by HPLC. The HPLC analyses were carried out on the Agilent technologies 1100 series equipped with an autosampler diode-array detector at 254 nm. The Zorbax SB C18 (15 cm × 4.6 mm i.d., 5 μm) reversed-phase column (Agilent technologies, USA) was run with a gradient solvent system initiated with 90% of solvent A (MeOH:water, 10:90, *v/v*) with 0.1% formic acid and 10% solvent B (MeOH with 0.1% formic acid) to 100% solvent A in 25 min. The flow rate was set at 1.0 mL/min. The column temperature was controlled at 40 °C. The quantitative data for daidzin, daidzein, genistin, and genistein were obtained from comparisons with known standards. Mass spectrometry was used to confirm the presence of substances in fermented soybeans with authentic standards. The MS analysis was carried out on with Agilent Technologies LC/MSD SL, Santa Clara, CA, USA. Capillary voltages of 4000 V (positive) and 3500 V (negative) were used with this analysis. The flow rate of N_2_ was set at 13 L/min at 320 °C on 60 psi of nebulizer pressure. The scan range of the mass spectral was 100–500 *m/z* in API-ES mode.

### 4.6. HPLC and LC-MS Analysis for Polyphenolic Compound Compositions

The polyphenolic compounds were separated according to the HPLC technique, with slight modifications [30]. The HPLC analyses were performed on the Agilent Technologies 1100 series, Germany, comprising a column oven (40 °C) and a diode-array detector at 270, 330, 350, and 370 nm. The column was a LiChroCART R-18e (15 cm × 4.6 mm i.d., 5 μm) reversed-phase column (Purospher STAR, Merck, Kenilworth, NJ, USA). The flow rate was 1.0 mL/min. The mobile phase was a binary solvent system consisting of (A) acetonitrile and (B) 10-mM ammonium formate buffer, pH 4 with formic acid, and the gradient used was 0–5 min B 100% constant, 5–10 min A 0–20%, 10–20 min A 20% constant, and 20–60 min A 20–40%. The compounds were identified by comparing with the standards of each identified compound using the retention time, the absorbance spectrum profile, and by running the samples after the addition of pure standards. The concentrations were calculated from the peak heights of the internal standard and each compound in the samples and reference solutions. Mass spectrometry was used to confirm the presence of substances in fermented soybeans with authentic standards. The MS analysis was performed on the Agilent Technologies LC/MSD SL, USA. Capillary voltages of 4000 V (positive) and 3500 V (negative) were used with this analysis. The flow rate of N_2_ was set at 13 L/min at 320 °C on 60 psi of nebulizer pressure. The scan range of the mass spectral was 100–700 *m*/*z* in API-ES mode.

### 4.7. Cell Culture

Professor Dr. Somdet Srichairatanakool, Department of Biochemistry, Faculty of Medicine, Chiang Mai University, Thailand generously provided human hepatocellular carcinoma (HepG2) cells, human embryonic kidney 293 (HEK293) cells, and human breast adenocarcinoma (MCF-7) cells. The HepG2 cells and HEK293 cells were cultured in 10% fetal bovine serum (FBS) in Dulbecco’s minimal essential medium (DMEM) supplemented with penicillin G (100 units/mL) and streptomycin (100 μg/mL) at 37 °C in a humidified atmosphere containing 5% CO_2_, while the MCF-7 cell was cultured in 10% fetal bovine serum (FBS) in Eagle’s MEM (EMEM) supplemented with penicillin G (100 units/mL) and streptomycin (100 μg/mL), nonessential amino acids (0.1 mM), insulin (10 µg/mL), sodium pyruvate (1 mM), and 10-nM estrogen at 37 °C in a humidified atmosphere containing 5% CO_2_.

### 4.8. Cytotoxic Analysis

The MTT (3-(4,5-dimethylthiazol-2-yl)-2,5-diphenyltetrazolium bromide) assay was used for the measurement of cell proliferation or when the metabolic events led to apoptosis or necrosis, a reduction in cell viability. The cells were treated as per the experimental design, and the incubation times were optimized for each cell type and system. The tetrazolium compound MTT was added to the wells, and the cells were incubated. The MTT was reduced by metabolically active cells to insoluble purple formazan dye crystals. Dimethyl sulfoxide (DMSO) was then added to the wells, solubilizing the crystals so the absorbance could be read using a spectrophotometer (BioTek Synergy H4 Hybrid Reader, BioTek Instruments Inc., Winooski, VT, USA). The samples were read directly in the wells. The wavelength for absorbance was 540 nm. The data were analyzed by plotting cell number versus concentrations of the compounds, allowing quantitation of the changes in cell proliferation. The rate of tetrazolium reduction was proportional to the rate of cell proliferation. After comparing the extract to the control, the percentage of inhibition was calculated, and the cytotoxicity of the extract was expressed as the concentration of the drug inhibiting the cell growth by 50% (IC_50_) [31]. The cells (HepG2 cells, HEK293 cells, and MCF-7 cells) were seeded into 96-well plates and treated with various concentrations of isoflavone (0–800 µg/mL), genistein, and daidzein (0–200 μM). Then, the treated cells were incubated at 37 °C for 24 and 48 h, and the cytotoxicity of the tested compounds was determined by using the MTT assay, as mentioned above.

### 4.9. Statistical Analysis

Results were expressed as the mean ± standard deviation (SD) from the three independent observations. Statistical analysis was performed using one-way analysis of variance (ANOVA) followed by Newman–Keuls using GraphPad Prism software (Version 7.00 for Windows, La Jolla, CA, USA). A *p*-value < 0.05 was considered statistically significant.

## 5. Conclusions

Our results showed that the isoflavone extract from Thua-nao displayed low inhibitory effects toward these cancer cell lines. However, due to the antioxidant properties of the extract and reported chemopreventive activities of the extract constituents, Thua-nao could exhibit chemopreventive properties, which will have to be further examined.

## Figures and Tables

**Figure 1 molecules-26-07432-f001:**
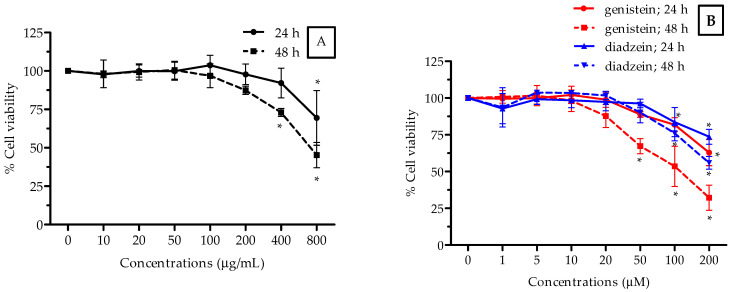
Cytotoxicity activity of ethanolic isoflavones extracts from Thua-nao (**A**) and genistein and diadzein (**B**) at various concentrations in HepG2 cells for 24 h and 48 h of incubation. All results are presented as the mean (*n =* 3) ± SD. * *p* < 0.05 compared to the nontreatment control at each incubation time.

**Figure 2 molecules-26-07432-f002:**
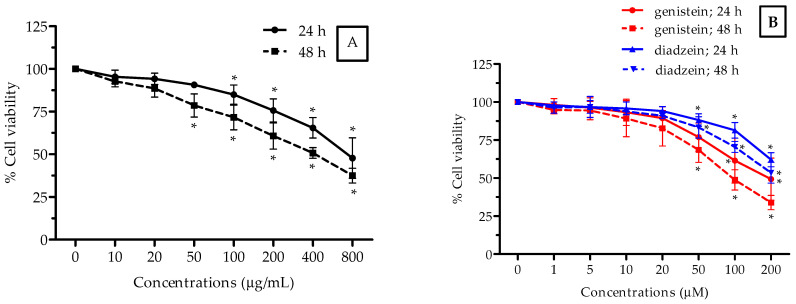
Cytotoxicity activity of the ethanolic isoflavone extracts from Thua-nao (**A**) and genistein and diadzein (**B**) at various concentrations in HEK293 cells for 24 h and 48 h of incubation. All results are presented as the mean (*n =* 3) ± SD. * *p* < 0.05 compared to the nontreatment control in each incubation time.

**Figure 3 molecules-26-07432-f003:**
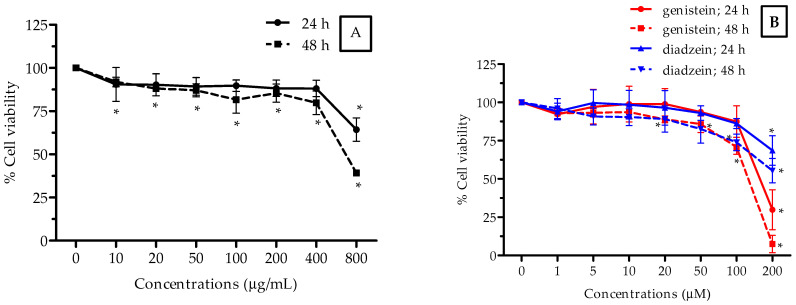
Cytotoxicity activity of the ethanolic isoflavone extracts from Thua-nao (**A**) and genistein and diadzein (**B**) at various concentrations in MCF-7 cells for 24 h and 48 h of incubation. All results are presented as the mean (*n =* 3) ± SD. * *p* < 0.05 compared to the nontreatment control in each incubation time.

**Table 1 molecules-26-07432-t001:** The total phenolic content, total flavonoid content, and antioxidant activities of ethanolic isoflavone extracts from Thua-nao.

	Ethanolic Isoflavones Extract
Total phenolic content (mg GAE/g of dry extract)	49.00 ± 0.51
Total flavonoid content (mg QE/g of dry extract)	10.76 ± 0.82
Antioxidant Activities	
ABTS (μmol Trolox/g of dry extract)	61.03 ± 0.97
FRAP (μM FeSO_4_/g of dry extract)	66.54 ± 3.97
DPPH (% inhibition)	22.47 ± 1.92

All data are expressed as the mean (*n* = 3) ± SD. GAE: gallic acid equivalent; QE: quercetin equivalent.

**Table 2 molecules-26-07432-t002:** The contents of the isoflavones and polyphenolic compounds of the ethanolic isoflavone extracts from Thua-nao.

	Amount (mg/kg of Dry Extract)
**Isoflavones**	
Genistein	616.80 ± 1.06
Daidzein	632.27 ± 1.11
Genistin	332.33 ± 1.53
Daidzin	251.39 ± 1.51
**Polyphenolic compounds**	
Gallic acid	65.84 ± 1.04
Eriodictyol	34.10 ± 1.02
Apigenin	<10.00
Isoquercetin	311.56 ± 1.50
Kaempferol	ND
Quercetin	98.89 ± 1.02
Hydroquinone	ND
Rutin	448.46 ± 1.50
Catechin	165.46 ± 1.50
Tannic acid	124.11 ± 1.02

All data are expressed as the mean (*n* = 3) ± SD. ND: not detected.

**Table 3 molecules-26-07432-t003:** Cytotoxic activity of ethanolic isoflavone extracts from Thua-nao in various cancer cells.

Types of Cancer Cell Line	IC_50_ (µg/mL)	
	24 h	48 h
HepG2	722	762
HEK293	716	392
MCF-7	698	>800

IC_50_: 50% inhibitory concentration; HepG2: human hepatocellular carcinoma cell line; HEK293: human embryonic kidney cell; MCF-7: human breast adenocarcinoma cell line.

## Data Availability

Data are available upon reasonable request.

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
