# Peer review of "Antioxidant Potential and Cytotoxic Effect of Isoflavones Extract from Thai Fermented Soybean (Thua-Nao)"

_molecules, 2021, doi:10.3390/molecules26247432_

Round 1

Reviewer 1 Report

The manuscript by Kulprachakarn et al. has been corrected regarding all points which I listed in previous review processes and now it is suitable to publish.

Author Response

Dear Reviewer,

Thank you for your kind acceptance.

Yours sincerely,

Kanokwan Kulprachakarn, Ph.D. 

Reviewer 2 Report

Dear Authors,

Manuscript ID:  molecules-1492563, Titled “Antioxidant Potential and Cytotoxic Effect of Isoflavones Extract from Thai Fermented Soybean (Thua-nao)” is improved.

Based on the evaluation of its originality, significance of content, scientific soundness, and interest to the readers, it may be considered for acceptance. Specific comments and a suggestion are provided below.

The Abstract is improved.

The Introduction is improved and the data on the antioxidant investigations on soybean are given as well as literature on techniques used for examination of its chemical compounds.

Material and methods are described properly in details.  

Results and discussion. The results are presented adequately.  The data for the antioxidant capacity (DPPH, ABTS) and FRAP activity are written in one subheading.

I should suggest Fig. 1 to be optimized and A, B and C graphics to be included in one graphic but in different colors/symbols; the same is recommended for Fig. 2.

In Discussion section, the obtained results are discussed and supported by literature data.

Author Response

Author’s response to Editorial Comments Form

Antioxidant Potential and Cytotoxic Effect of Isoflavones Extract from Thai Fermented Soybean (Thua-nao) (Manuscript ID: molecules- 1492563)

Thank you very much for your valuable comments and also for giving us a chance to improve our manuscript. We have responded and amended in comments (in blue highlight). Please find our comments and responses below. If you need any further information, please do not hesitate to let me know.

Yours sincerely,

Kanokwan Kulprachakarn, Ph.D.     

…………………………………………………………………………………………………

#Reviewer 2

Manuscript ID:  molecules-1492563, Titled “Antioxidant Potential and Cytotoxic Effect of Isoflavones Extract from Thai Fermented Soybean (Thua-nao)” is improved.

Based on the evaluation of its originality, significance of content, scientific soundness, and interest to the readers, it may be considered for acceptance. Specific comments and a suggestion are provided below.

  1. The Abstractis improved.

Answer: Thank you for your comment.

  1. The Introductionis improved and the data on the antioxidant investigations on soybean are given as well as literature on techniques used for examination of its chemical compounds.

Answer: Thank you for your comment.

  1. Material and methods are described properly in details.  

Answer: Thank you for your comment.

  1. Results and discussion. The results are presented adequately.  The data for the antioxidant capacity (DPPH, ABTS) and FRAP activity are written in one subheading.

Answer: Thank you for your comment.

  1. I should suggest Fig. 1 to be optimized and A, B and C graphics to be included in one graphic but in different colors/symbols; the same is recommended for Fig. 2.

Answer: Thank you for your kind suggestion. We have edited Figures 1, 2, and 3 by combining (B) and (C) in one graphic in the Results part (page 6-9). Due to the difference of unit and scale of the concentration in graphic (A) is µg/mL whereas (B) and (C) are µM. Thus, we cannot optimize in one graphic.

  1. In Discussion section, the obtained results are discussed and supported by literature data.

 Answer: Thank you for your comment.

…………………………………………………………………………………………………..

Yours sincerely,

Authors

Reviewer 3 Report

The authors have made improvements to the manuscript. However, the conclusions are still not in line with the obtained results. The authors mention in the abstract that the extract inhibits cancer cell growth. However, this effect is not significant, with IC50 values ranging from 400 to > 800 µg/ml.  As previously suggested, the authors mention that the extract could have chemopreventive properties due to its free-radical scavenging properties. However, this chemopreventive activity of Thua-nao is not backed by the results or discussed. I suggest, apart from displaying the protective role of Thua-nao, the authors could elaborate in the discussion on the known chemopreventive mechanisms of the active constituents, genistein and daidzein.

Author Response

Author’s response to Editorial Comments Form

Antioxidant Potential and Cytotoxic Effect of Isoflavones Extract from Thai Fermented Soybean (Thua-nao) (Manuscript ID: molecules- 1492563)

Thank you very much for your valuable comments and also for giving us a chance to improve our manuscript. We have responded and amended in comments (in blue highlight). Please find our comments and responses below. If you need any further information, please do not hesitate to let me know.

Yours sincerely,

Kanokwan Kulprachakarn, Ph.D.     

…………………………………………………………………………………………………

#Reviewer 3

  1. The authors have made improvements to the manuscript. However, the conclusions are still not in line with the obtained results. The authors mention in the abstract that the extract inhibits cancer cell growth. However, this effect is not significant, with IC50 values ranging from 400 to > 800 µg/ml.

Answer: Thank you for your comment. We have rewritten the sentence in both the Abstract and Discussion parts as following “Isoflavones demonstrated the weak inhibition on MCF-7 and HEK293 cancer cells growth.” (page 1) and “Our results showed that the isoflavones extract from Thua-nao have a low inhibitory effect on these cancer cell line growth but this effect was not significant.” (page 11)

  1. As previously suggested, the authors mention that the extract could have chemopreventive properties due to its free-radical scavenging properties. However, this chemopreventive activity of Thua-nao is not backed by the results or discussed. I suggest, apart from displaying the protective role of Thua-nao, the authors could elaborate in the discussion on the known chemopreventive mechanisms of the active constituents, genistein and daidzein.

Answer: Thank you for your kind suggestion. We have amended in the Discussion part (page 11) as following “Genistein and daidzein are two of the most important active compounds found in soybeans, both of which play a vital role in cancer prevention [22]. Genistein has been shown to inhibit cell growth in a variety of cultured cancer cells, including breast, lung, and prostate cancers, as well as leukemia and lymphoma [14,23]. It affects the cell cycle, apoptosis, angiogenesis, and metastasis at various stages. Remarkably, genistein is considered to have anticancer properties through a variety of mechanisms, including tyrosine-specific protein kinases [5], caspases, B-cell lymphoma 2 (Bcl-2), Bcl-2-associated X protein (Bax), mitogen-activated protein kinase (MAPK), nuclear factor-κB (NF-κB), inhibitor of NF-κB, extracellular signal-regulated kinase 1/2 (ERK 1/2), phosphoinositide 3-kinase/Akt (PI3K/Akt), and Wingless and integration 1/β-catenin (Wnt/β-catenin) signaling pathway [24].”

…………………………………………………………………………………………………..

Yours sincerely,

Authors

Round 2

Reviewer 3 Report

The authors have addressed my comments. I have some additional recommendations:

I suggest changing the conclusions (line 250) to:  

Our results showed that the isoflavone extract from Thua-nao displayed low inhibitory effects toward these cancer cell lines. However, due to the antioxidant properties of the extract and reported chemopreventive activities of the extract constituents, Thua-nao could exhibit chemopreventive properties, which would have to be further examined.

The manuscript requires language corrections.

Author Response

Author’s response to Editorial Comments Form

Antioxidant Potential and Cytotoxic Effect of Isoflavones Extract from Thai Fermented Soybean (Thua-nao) (Manuscript ID: molecules- 1492563)

Thank you very much for your valuable comments and also for giving us a chance to improve our manuscript. We have responded and amended in comments (in green highlight). Please find our comments and responses below. If you need any further information, please do not hesitate to let me know.

Yours sincerely,

Kanokwan Kulprachakarn, Ph.D.

…………………………………………………………………………………………………

#Reviewer 3

The authors have addressed my comments. I have some additional recommendations:

  1. I suggest changing the conclusions (line 250) to:

Our results showed that the isoflavone extract from Thua-nao displayed low inhibitory effects toward these cancer cell lines. However, due to the antioxidant properties of the extract and reported chemopreventive activities of the extract constituents, Thua-nao could exhibit chemopreventive properties, which would have to be further examined.

Answer: Thank you for your valuable suggestion. We have amended and rewritten in the Conclusion part (page 12).

  1. The manuscript requires language corrections.

Answer: Thank you for your precise comment. We have checked and corrected all manuscript.

…………………………………………………………………………………………………

Yours sincerely,

Authors

This manuscript is a resubmission of an earlier submission. The following is a list of the peer review reports and author responses from that submission.

Round 1

Reviewer 1 Report

The manuscript is methodologically supplemented but there are still oversights. Namely, you added two antioxidant tests and kept only results from ABTS in the abstract without precise what it is reflected. Line 203- you cited previously founds concerning DPPH test, but did not mention the results expression way?

It seems authors are not familiar with terminology in the research filed. This can be seen from the addition- lines 220-221 ("Interestingly, it was also reported that the aglycone forms of daidzein and genistein have the highest isoflavone content.")

Authors stated that they had amended the wavelength for ABTS assay, but results stayed absolutely the same. Just replaced 517 to 734nm?! It is not practically achievable.

Reviewer 2 Report

Dear Authors,

Manuscript ID:  molecules-1472195, Titled “Antioxidant Potential and Cytotoxic Effect of Isoflavones Extract from Thai Fermented Soybean (Thua-nao)” is well-structured.

Based on the evaluation of its originality, significance of content, scientific soundness, and interest to the readers, a minor revision is suggested before the article may be considered for acceptance. Specific suggestions, recommendations and comments are provided below.

The title is concise and declares the object of the research.

The Abstract is factual and well-structured. It states briefly the aim of the study, the used methods and the major results. The conclusion suggests further future investigations.

The Introduction presents data concerning the Thai fermented soybean (Thua-nao); its traditional use in Asian countries; the secondary biological chemicals known as isoflavones which are found in soybeans and their effect on different physiological disorders and diseases.

The literature survey is precise. Nevertheless, data for antioxidant investigations on soybean are not given as well as literature on techniques used for examination of its chemical compounds e.g.

HPLC and LC-MS techniques. The above-mentioned should be added.

Material and methods are described in details, divided with subheadings according to the procedure. The assays for examining the radical scavenging activity of Thua-nao as DPPH, ABTS and ferric (FRAP) reducing power are done to complete the study.  

Results and discussion. The results are presented adequately. However, the data written in the text for the antioxidant capacity (DPPH, ABTS) and FRAP activity repeat directly the data from Table 1 (Lines 108-124) and it should be avoided. The information how ABTS radical scavenging and FRAP activity results were expressed is not appropriate here (Line 110 and Lines 116-117, respectively) since it is given in Material and Methods section.  It should be taken in mind.

I should suppose the antioxidant activity results (DPPH, ABTS and FRAP) to be not divided by subheadings but written in one.

Figure 2 should be optimized and I recommend the A, B and C to be included in one graphic in different colors/symbols.

The discussion explores the main results of the work. However, the obtained data are barely discussed and explained. Discussion is written with a lot of data and general information, which is more appropriate for the Introduction section. Based to the aforementioned, Discussion section should be improved.  

The Conclusion supposed a subsequent investigation on the benefits of Thua-nao.

Reviewer 3 Report

The article presents research regarding the antioxidant activity and cytotoxic potential of a Thai fermented soybean extract. Below are points to be addressed by the authors:

The research lacks a discussion comparing the antioxidant potential of the examined Thua-nao extract in comparison to other Thua-nao preparations or fermented soybean foods, as the mentioned novelty of this current study is related to the Bacillus strains used for the preparation of Thua-nao. Therefore, some conclusions as to the potential of the presented Thua-nao preparation would be beneficial.

There are some discrepancies in this study. The authors mention that the novelty of this study relates to the use of a Bacillus subtilis var. Thua-nao strain. This is not in line with what is mentioned in the Materials and Methods section.

The authors mention ‘the aglycone forms of daidzein and genistein have the highest isoflavone content’ this should be rephrased. Furthermore, the authors could discuss the isoflavone content of the obtained extract in relation to other fermented soybean foods.

The authors mention that the extract inhibited cancer cell growth, which is not supported by the results. Furthermore, the authors mention that the extract could protect against breast cancer development due to the antioxidant and anti-inflammatory potential of the present constituents. This direction could have been undertaken in this research and the protective effects of this extract could have been determined, which would add novelty to this study. For example the free-radical scavenging properties of the extract could be supported by studies showing the protective role of the extract against ROS-inducing agents in MCF-7 cells.